# Contextualising COVID-19 prevention behaviour over time in Australia: Patterns and long-term predictors from April to July 2020 in an online social media sample

**Julie Ayre**◉\*, **Erin Cvejic**◉, **Kirsten McCaffery**◉, **Tessa Copp, Samuel Cornell, Rachael H. Dodd**◉**, Kristen Pickles, Carys Batcup, Jennifer M. J. Isautier, Brooke Nickel**◉**, Thomas Dakin, Carissa Bonner**

Sydney Health Literacy Lab, School of Public Health, Faculty of Medicine and Health, The University of Sydney, Sydney, New South Wales, Australia

\* Julie.ayre@sydney.edy.au

## Abstract

### Background

In Australia in March 2020 a national public health directive required that non-essential workers stay at home, except for essential activities. These restrictions began easing in May 2020 as community transmission slowed.

### Objectives

This study investigated changes in COVID prevention behaviours from April-July 2020, and psychosocial predictors of these behaviours.

### Methods

An Australia-wide (national) survey was conducted in April, with monthly follow-up over four months. Participants who were adults (18+ years), currently residing in Australia and who could read and understand English were eligible. Recruitment was via online social media. Analysis sample included those who provided responses to the baseline survey (April) and at least one subsequent follow-up survey (N = 1834 out of a possible 3216 who completed the April survey). 71.7% of the sample was female (n = 1,322). Principal components analysis (PCA) combined self-reported adherence across seven prevention behaviours. PCA identified two behaviour types: 'distancing' (e.g. staying 1.5m away) and 'hygiene' (e.g. washing hands), explaining 28.3% and 24.2% of variance, respectively. Distancing and hygiene behaviours were analysed individually using multivariable regression models.

### Results

On average, participants agreed with statements of adherence for all behaviours (means all above 4 out of 7). Distancing behaviours declined each month (p's < .001), whereas hygiene behaviours remained relatively stable. For distancing, stronger perceptions of societal risk,

**Data Availability Statement:** All relevant data are within the manuscript and its Supporting Information files.

**Funding:** The authors received no specific funding for this work.

**Competing interests:** The authors have declared that no competing interests exist.

self-efficacy to maintain distancing, and greater perceived social obligation at baseline were associated with adherence in June and July (p's<0.05). For hygiene, the only significant correlate of adherence in June and July was belief that one's actions could prevent infection of family members (p < .001).

## Conclusion

High adherence to COVID prevention behaviours were reported in this social media sample; however, distancing behaviours tended to decrease over time. Belief in social responsibility may be an important aspect to consider in encouraging distancing behaviours. These findings have implications for managing a shift from government-imposed restrictions to individual responsibility.

## Introduction

COVID-19 has already had a huge global impact on health, mortality and economies. In lieu of a vaccine or effective treatment, public health policies initially focused on managing the spread of the virus so that health needs do not surpass health system capacity [1, 2]. In Australia, key policies have involved restricting international and interstate travel (including a 14-day hotel quarantine), restricting how often and for what reasons people are able to leave their homes, and intensive contact tracing of individuals who test positive. Some restrictions and prevention behaviours have been enforced through penalties, fines or administrative processes; others are encouraged but compliance is ultimately up to the individual. For example, the Australian public health response to COVID-19 has included distance from home limits and fines in outbreak areas, while strongly encouraging hygiene practices such as washing hands and getting tested if even mild COVID-19 symptoms occur (e.g. cough, fever, sore throat).

Encouraging individual uptake of COVID prevention behaviours without regulation presents an important challenge for public health. Whilst environmental cues and other external factors can increase the likelihood of these behaviours, they may not be sufficient for sustained uptake. For example, coloured markers on the ground can be used to indicate appropriate spacing in queues, but adherence to this advice is ultimately at the discretion of the individual. Similarly, although many restaurants and cafés provide hand sanitiser, it would be impractical to enforce their uptake.

Despite this challenge, Australian community surveys have suggested high public support for COVID prevention behaviours. For example, a series of five nationally representative cross-sectional surveys from April to August 2020 indicate high levels of physical distancing ('keeping distance from people' endorsed by over 90% at each time-point) [3–7]. These surveys provide useful point estimates of the prevalence of COVID prevention behaviours. However, participants were not followed over time, and not all behaviours were measured at each time point. As such, we have limited understanding of how these behaviours may have changed over time.

The cross-sectional nature of the data also limits our ability to gauge how psychosocial factors influence engagement in COVID prevention behaviours. The broader behaviour change literature and theoretical models suggest that self-efficacy (confidence to perform a behaviour), attitudes towards the target behaviour, social norms and risk perception, often play an important role in uptake of health behaviours [8–11]. However, to date, only a few peer-reviewed studies in Australia, the United Kingdom, and the United States have investigated whether

psychosocial factors tied to behavioural theories can predict continued adherence to COVID prevention behaviours [12–14]. For example, three studies reported that intention and confidence to perform/maintain a COVID prevention behaviour (self-efficacy) were consistently associated with self-reported behaviour one week later [12–14]. Two of these studies also measured action planning and self-monitoring and observed that these constructs predicted distancing behaviour [12, 13]. In contrast, there is mixed evidence that risk perception, social and moral norms, and belief in the behaviour's utility contribute to distancing behaviour, although these are often reported as correlates of COVID prevention behaviours in cross-sectional studies [15–21].

These findings provide valuable insight into which psychosocial factors may be the most influential drivers of COVID prevention behaviours. In doing so, these studies could help identify strategies to improve the effectiveness of behavioural interventions. However, it should be noted that these studies were conducted over a very short period of time (1 week) so findings may not generalise to longer-term maintenance of these behaviours. The short follow-up period might also mean that findings are tied to the specific context at that point in time. For example, each of the longitudinal studies described above were conducted in April or May 2020 [12–14]. During this time in Australia, national restrictions limited the extent that people could leave their homes. In subsequent months these restrictions have eased, presenting new challenges as people have adapted to the responsibility to regulate their own social distancing in public domains (e.g. restaurants, gyms, and on public transport). Understanding how people have responded to COVID-19 restrictions and guidelines over time is a pressing and controversial issue. There has been much debate and political discourse about the concept of 'behavioural fatigue,' [22–24] but reduced behavioural adherence in Australia may be the result of changing COVID-19 circumstances such as low community transmission and easing restrictions, rather than mental difficulty or tiredness in sustaining behaviours.

This study seeks to address this gap in the literature by investigating seven key COVID prevention behaviours and several potential important psychosocial constructs, using a longitudinal design that extends over a four-month period. These behaviours include staying at home, keeping 1.5 metres away from others, washing hands frequently, avoiding touching one's face, avoiding close contact with people with symptoms, cleaning frequently touched surfaces, and avoiding shaking hands, hugging or kissing as a greeting. The time periods investigated in this study include April and May (a period of strong national restrictions), June (fewer restrictions, and a low and stable number of daily COVID-19 cases), and July (a period with localised outbreaks), 2020 (Fig 1; explained in greater detail in the 'Setting' section of the Methods).

Specifically, this study aims to: 1) investigate patterns of COVID prevention behaviours in an Australian sample (April to July 2020); and 2) investigate (April) psychosocial predictors of COVID prevention behaviours in June and July 2020.

## Materials and methods

### Design

The data used in this study are from a prospective longitudinal national survey launched in Australia exploring variation in understanding, attitudes, and uptake of COVID-19 health advice during the 2020 pandemic [25]. The baseline survey was launched in April (17th– 24th), around one month after national restrictions and hotel quarantine procedures were introduced for international travellers. A subset of participants (n = 3214) were invited to complete surveys during May 8 – 15th, June 5 – 12th, and July 23rd– 31st.

Participants were eligible if they were aged 18 years and over, could read and understand English and were currently residing in Australia. Recruitment was via paid advertisements on

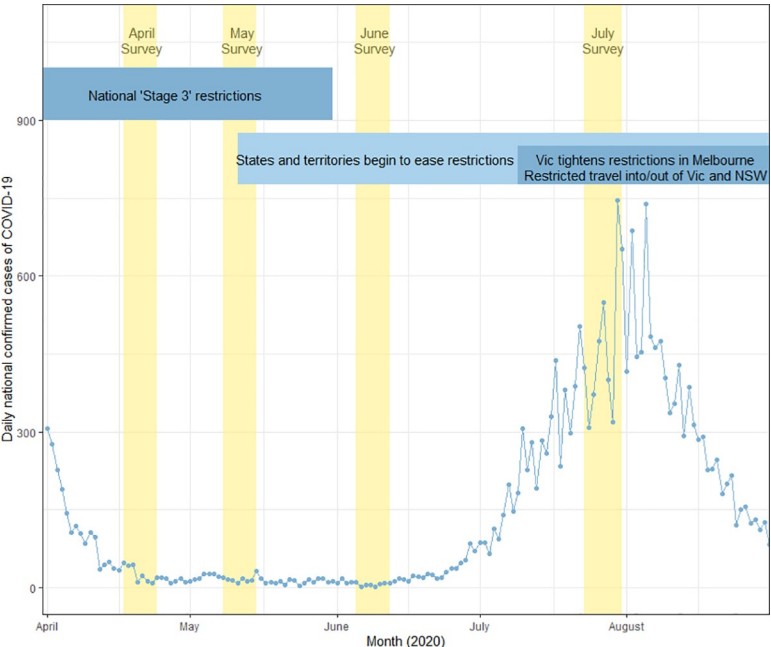

**Fig 1. Summary of daily national confirmed cases of COVID-19, survey timing and government COVID-19 restrictions in Australia, April to August 2020.** Vertical yellow bars indicate timespans for Surveys 1 to 4. NSW: New South Wales; VIC; Victoria. Source for national case data: covid19data.com.au.

Facebook and Instagram and the survey was hosted by Qualtrics, an online survey platform. Participants were given the opportunity to enter into a prize draw for the chance to win one of ten $20 gift cards upon completion of each survey wave. This study was approved by The University of Sydney Human Research Ethics Committee (2020/212).

## Setting

In late March 2020, Australia entered 'Stage 3' restrictions. That is, all non-essential workers were required to work from home, no guests could enter another household, gatherings were limited to two adults (unless from the same household), and non-essential services were closed. This was enforced in some states using penalties and fines. There were small differences across the states and territories. On May 8[th], the federal government released a pathway for states and territories to ease restrictions. Most of the states and territories began easing restrictions in the second week of May (11[th]-17[th]) [26]. Western Australia had already begun easing restrictions at the end of April. Initially, Australians were only able to test for COVID-19 if they were at higher risk of infection [27]. From April, the states and territories began to remove restrictions and encouraged anyone with symptoms to get tested, however mild (e.g. cough, fever, sore throat). Following an outbreak in Melbourne, Victoria, restrictions in Melbourne were tightened on July 9[th], this time to a greater degree than the national restrictions implemented in March (e.g. by imposing a curfew). These tighter restrictions were still in place at the time of the July survey. Some additional smaller outbreaks also appeared in July in NSW.

## Measures

Age, gender, education, language other than English (LOTE) spoken at home, and socioeconomic status (SES) were assessed at baseline in April as detailed in [25]. Seven COVID

prevention behaviours were assessed by asking participants 'Please tell us what you have been doing over the last 4 weeks. Mark how much you agree or disagree with the following statements.' Responses were recorded using a 7-point Likert scale anchored 'strongly disagree' to 'strongly agree.' These statements described seven COVID prevention behaviours: staying at home, keeping 1.5 metres away from others, washing hands frequently, avoiding touching one's face, avoiding close contact with people with symptoms, cleaning frequently touched surfaces, and avoiding shaking hands, hugging or kissing as a greeting (exact wording shown in Table 1). Perceived societal risk of COVID-19 was assessed using a 10-point Likert scale ('no threat at all' to 'a very serious public health threat'); personal risk perception was assessed by the item: 'Do you think that you will get sick from COVID-19?' (dichotomised in this study as 'Not at all' vs. 'It's possible' or greater) [28]. Self-efficacy was assessed by asking about the extent that participants felt they could maintain social distancing for the next month, or the next three months (7-point Likert scales). Beliefs in the consequences of personal actions was assessed both in terms of ability to prevent oneself from contracting the virus (10-point Likert scale) [28] and prevent one's family from contracting the virus (7-point Likert scale) [29]. Social obligation was assessed using three items adapted from a collective responsibility subscale [30] and one additional item about following government public health guidelines. Other relevant measures and individual items are displayed in Table 1.

## Statistical analysis

Statistical analyses were conducted using Stata/IC v16.1 (StataCorp, College Station, TX, USA). The analysis sample comprised participants from our prospective longitudinal study [25] who provided responses to the baseline survey (April) and at least one subsequent follow-up survey (N = 1,843). No new participants were added at subsequent time points. Participants in the analysis sample were similar to those of the full sample, though a higher proportion were female (71.7% vs 63.5% female) and older (27.1% aged 56 to 90 years vs 23.3%). A comparison of the full and analysed sample is provided in S1 Table. Descriptive statistics (means and standard deviations for continuous variables, frequencies and relative frequencies for categorical variables) for the participants characteristics were generated, and compared with ABS data were possible, and for study outcomes (Table 2). Principal components analysis (PCA) with varimax rotation was used to reduce the dimensionality of participant responses to COVID prevention behaviours at baseline (April survey), with the component loadings re-projected onto responses provided in subsequent surveys. Linear mixed models (with random intercepts by participant) were used to examine the temporal variations of the extracted components. Of note, the public health directive to stay at home except for essentials was relaxed in most states by the time of the June and July surveys. For this reason, we also conducted a sensitivity analysis that excluded this behaviour from the model. Associations of the extracted components for the June and July surveys, with a range of personal and societal risk perceptions, self-efficacy, personal control, social obligation beliefs, and sociodemographic factors, were explored in a series of multivariable linear regression models. As these analyses were exploratory and hypothesis-generating, a p-value of .05 was set as the threshold for statistical significance.

## Results

### Sample characteristics

1,843 participants were included in the analysis sample. Median age of the sample was 42 years, 71.7% of the sample was female (n = 1,322), and 73.1% of the sample had a university

**Table 1. Items used to assess COVID prevention behaviours and related psychosocial factors.**

| Construct | Survey item | Response options |
|---|---|---|
| **COVID prevention behaviour** | I stay at home unless I need to shop for food or medicine, exercise, go to work, or provide care/support to another | 1 (strongly disagree) to 7 (strongly agree) |
| | I wash my hands frequently with soap and water (for at least 20 seconds). For example, before and after eating, after going to the toilet, and after going outside | |
| | I stay 1.5m away from other people outside my home | |
| | I avoid close contact with anyone with cold or flu like symptoms | |
| | I avoid touching my eyes, nose and mouth with unwashed hands | |
| | I clean and disinfect frequently touched surfaces each day (e.g. phones, keyboards, door handles, light switches, bench tops) | |
| | I have stopped shaking hands, hugging or kissing as a greeting | |
| **Societal risk perception** | On a scale of 1 to 10, how serious of a public health threat do you think COVID-19 is currently? * ('or will become' for wave 1 version) | 1 (no threat at all) to 10 (a very serious public health threat) |
| **Personal risk perception** | Do you think that you will get sick from COVID-19? | 1: Not at all |
| | | 2: It's possible |
| | | 3: I probably will |
| | | 4: I definitely will |
| **Self-efficacy to maintain distancing** | I will find it very hard to follow social distancing for the next month if this is recommended | 1 (strongly disagree) to 7 (strongly agree) |
| | I will find it very hard to follow social distancing for the next 3 months if this is recommended | |
| **Consequence of personal actions** (Control over likelihood of personal/ family COVID-19 infection) | My actions will influence whether or not I get COVID-19 | 1 (do not agree at all) to 10 (agree very strongly) |
| | Social distancing is important for my family's health | 1 (strongly disagree) to 7 (strongly agree) |
| **Social obligation** | It is my responsibility to follow all public health guidance to prevent the spread of COVID-19 to others* | 1 (strongly disagree) to 7 (strongly agree) |
| | Social distancing is important for the health of others in my community | |
| | When everyone else is socially distancing, I don't need to (reverse coded) | |
| | I socially distance to protect people with a weaker immune system | |

* Participants indicated responses to 'It is my responsibility to follow all public health guidance to prevent the spread of COVID-19 to others,' using a 10-point Likert scale ('do not agree at all' to 'agree very strongly'). Response were rescaled in order to combine this item with the remaining social obligation items.

level of education (n = 1,347). Comparison to national population estimates is available in Table 2.

On average, scores for each of the seven COVID prevention behaviours at baseline (April) were above the mid-point of the scale (i.e. 4 out of 7), indicating agreement with statements of

**Table 2. Descriptive characteristics of analysis sample at baseline (April) (N = 1843).**

| Characteristic | | Analysis sample (N = 1843) n (%) | National Australian estimates |
|---|---|---|---|
| Age, median (IQR) | | 42 (28–57) | |
| Age group | | | |
| | 18 to 25 years | 353 (19.2%) | 13.7% |
| | 26 to 40 years | 528 (28.6%) | 27.2% |
| | 41 to 55 years | 462 (25.1%) | 26.0% |
| | 56 to 90 years | 500 (27.1%) | 33.1% |
| Gender | | | |
| | Male | 487 (26.4%) | 48.9% |
| | Female | 1322 (71.7%) | 51.1% |
| | Other/prefer not to say | 34 (1.8%) | - |
| Educational attainment* | | | |
| | Less than university | 496 (26.9%) | 53.7% |
| | University | 1347(73.1%) | 32.4% |
| State/territory of residence | | | |
| | Australian Capital Territory | 58 (3.1%) | 1.7% |
| | Northern Territory | 7 (0.4%) | 0.9% |
| | Victoria | 291 (15.8%) | 25.5% |
| | New South Wales | 937 (50.8%) | 32.0% |
| | Queensland | 258 (14.0%) | 19.9% |
| | Western Australia | 130 (7.1%) | 10.5% |
| | South Australia | 84 (4.6%) | 7.3% |
| | Tasmania | 78 (4.2%) | 2.2% |
| Residential area remoteness^ | | | |
| | Major cities | 1374 (74.6%) | 73.3% |
| | Regional and remote | 467 (25.4%) | 26.5% |
| Socioeconomic status, mean IRSAD quintile (SD) [†] | | 3.66 (1.40) | |
| Born in Australia | | 1405 (76.2%) | 61.3% |
| English primary language | | 1774 (96.3%) | 71.5% |
| Aboriginal or Torres Strait Islander | | | |
| | Yes | 31 (1.7%) | 2.1% |
| | No | 1796 (97.4%) | 91.6% |
| | Did not respond | 16 (0.9%) | 6.2% |
| Chronic health conditions** | | | |
| | None | 912 (49.5%) | 52.7% |
| | One | 537 (29.1%) | 27.0% |
| | Two or more | 394 (21.4%) | 20.2% |
| Health literacy adequacy^^ | | 1695 (92.0%) | - |
| Self-Reported General Health[††] | | | |
| | Poor | 68 (3.7%) | 3.9% |
| | Fair | 248 (13.5%) | 11.3% |
| | Good | 629 (34.1%) | 29.1% |
| | Very good | 667 (36.2%) | 35.4% |
| | Excellent | 231 (12.5%) | 20.2% |

National Australian estimates are based on 2016 Australian census data, for people aged 19–90 years, except where indicated.

*13.8% of census data listed supplementary codes or 'not stated' for highest level of educational attainment.

^Remoteness indicators are based on 2016 ABS data, and as such, individuals who reside in newer postcodes established after 2016 (n = 2) are missing data on this variable.

[†]Quintile 1 indicates a participant resides in one of the least advantaged (most disadvantaged) areas; quintile 5 indicates a participant resides in on the most advantaged (least disadvantaged) areas.

**National estimates based on 2017–18 National Health Survey, people aged ≥15 years.

Chronic conditions included arthritis, asthma, back problems, cancer, chronic obstructive pulmonary disease (COPD), diabetes mellitus, heart, stroke and vascular disease, kidney disease, mental and behavioural conditions and osteoporosis.

^^Based on Single Item Literacy Screener (SILS): How confident are you with filling out medical forms by yourself: not at all, a little bit, somewhat, quite a bit, extremely. "Not at all" response categorised as inadequate health literacy.

[††]National estimates based on 2017–18 National Health Survey, people aged ≥18 years.

**Table 3. Descriptive statistics of individual COVID-19 protection behaviours and relevant psychosocial factors, across four survey waves.**

| | April survey (n = 1843) | May Survey (n = 1649) | June Survey (n = 1206) | July Survey (n = 1126) | Change* |
|---|---|---|---|---|---|
| **COVID-19 behaviours: 1 (low) to 7 (high)** | | | | | |
| Staying at home | 6.5 (1.0) | 6.3 (1.1) | 5.6 (1.6) | 5.0 (1.8) | 1.51 |
| Avoiding shaking hands, hugging or kissing as a greeting | 6.6 (0.9) | 6.5 (1.0) | 6.1 (1.3) | 6.0 (1.3) | 0.64 |
| Staying 1.5m away from other people | 6.3 (1.0) | 6.2 (1.1) | 5.9 (1.2) | 5.9 (1.1) | 0.42 |
| Washing hands frequently | 6.2 (1.1) | 6.1 (1.2) | 6.0 (1.2) | 6.0 (1.1) | 0.29 |
| Avoiding touching face | 5.6 (1.4) | 5.4 (1.4) | 5.4 (1.4) | 5.4 (1.3) | 0.20 |
| Avoiding close contact with anyone with cold or flu like symptoms | 6.5 (0.9) | 6.5 (1.0) | 6.4 (0.9) | 6.4 (1.0) | 0.16 |
| Cleaning frequently touched surfaces | 4.3 (1.9) | 4.2 (1.8) | 4.2 (1.8) | 4.3 (1.8) | -0.05 |
| **Risk perception** | | | | | |
| Societal risk perception^: 1 (low risk) to 10 (high risk) | 7.9 (2.1) | 6.3 (2.3) | 5.6 (2.4) | 8.0 (1.9) | |
| Personal risk perception (not at all likely to get sick with COVID-19; n, %) | 68 (3.8) | 94 (5.7) | 106 (8.8) | 37 (3.3) | |
| **Self-efficacy to maintain distancing** | | | | | |
| Over a 1-month period: 1 (low) to 7 (high) | 5.4 (1.7) | 5.4 (1.7) | *N/A* | *N/A* | |
| Over a 3-month period: 1 (low) to 7 (high) | 4.7 (2.0) | 4.8 (2.0) | *N/A* | *N/A* | |
| **Perceived consequences of personal actions** | | | | | |
| Control over likelihood of personal COVID-19 infection: 1 (low) to 10 (high) | 8.4 (1.8) | 8.2 (1.8) | 8.1 (1.8) | 8.2 (1.7) | |
| Control over likelihood of family COVID-19 infection: 1 (low) to 7 (high) | 6.5 (0.9) | 6.4 (0.9) | 6.3 (1.0) | 6.4 (0.9) | |
| **Social obligation** | | | | | |
| Social obligation: 1 (low) to 7 (high) | 6.6 (0.6) | 6.5 (0.6) | 6.5 (0.7) | 6.5 (0.7) | |

Data are displayed as means (standard deviations) unless otherwise specified.

^ in the April survey, this question was given without locational framing; for the remaining surveys this was asked specifically within an Australian context. N/A indicates this question was not assessed.

*Difference scores are for participants who provided responses for the April and July surveys (n = 1,126).

adherence. The mostly strongly endorsed statement was 'I have stopped shaking hands, hugging or kissing as a greeting' (M = 6.6 out of 7, SD = 0.9), followed by, 'I avoid close contact with anyone with cold or flu like symptoms' (M = 6.5, SD = 0.9), and 'I stay at home unless I need to shop for food or medicine, exercise, go to work, or provide care/support to another' (M = 6.5, SD = 1.0). The statement with weakest endorsement was 'I clean and disinfect frequently touched surfaces each day' (M = 4.3 out of 7, SD = 1.9). For subsequent surveys, mean scores reduced for some behaviours but all remained above the mid-point of the scale (Table 3).

In April, participants perceived COVID-19 as a relatively serious public health threat, rating societal risk on average 7.9 out of 10 (SD = 2.1). Average perceived societal risk reduced to 5.6 in June (SD = 2.4), and returned to baseline levels of 8.0 in July (SD = 1.9), coinciding with the COVID-19 outbreak in Victoria. Perceptions of personal risk were similar; only 3.8% of participants (n = 68) felt they were 'not at all' at risk of COVID-19 infection in April; this proportion was 8.8% (n = 106) in June and 3.3% (n = 37) in July. These fluctuations mirror the change in daily national COVID-19 cases at the time of each survey, with higher community transmission in April and July, and lower case numbers in June (Fig 1). On average across each survey, participants reported high self-efficacy to maintain distancing behaviours (means ranged from 4.7 to 5.4 out of 7), strong perceived ability to prevent infection for themselves (all means

**Table 4. Rotated component loadings from PCA for COVID prevention behaviours at baseline (April survey).**

|  | Component 1: Distancing | Component 2: Hygiene |
|---|---|---|
| Staying at home | **0.5744** | -0.1601 |
| Washing hands frequently | 0.2472 | **0.3773** |
| staying 1.5m away from other people | **0.5434** | -0.0505 |
| Avoiding close contact with anyone with cold or flu like symptoms | **0.3847** | 0.1076 |
| Avoiding touching face | 0.0750 | **0.5750** |
| Cleaning frequently touched surfaces | -0.1501 | **0.6875** |
| Avoiding shaking hands, hugging or kissing as a greeting | **0.3708** | 0.1209 |

Loadings > |0.3| shown in bold.

above 8 out of 10) or family members (all means above 6 out of 7), and social obligation to adhere to COVID prevention behaviours (all means above 6 out of 7) (Table 3).

## Relationship between COVID prevention behaviours

The level of agreement between the seven COVID prevention behaviours during April indicated moderate internal consistency (Cronbach's α = 0.72), and sufficient sampling adequacy (KMO = 0.78). Application of PCA with varimax rotation identified a two-component solution with eigenvalues greater than 1, cumulatively accounting for 52.5% of the variance (see Table 4 for rotated component loadings). Examination of the contributing items to each component resulted in the labels "distancing" (component 1; accounting for 28.3% of the variance) and "hygiene" (component 2; accounting for 24.2% of the variance) being applied. Component loadings were then reprojected onto responses for subsequent survey timepoints. Pairwise correlations between the distancing and hygiene components (herein referred to as distancing and hygiene 'behaviours') were similar across time (April Survey: r = 0.44, p < .001; May Survey: r = 0.45, p < .001; June Survey: r = 0.47, p < .001; July Survey: r = 0.45, p < .001).

Linear mixed models analysis (with random intercepts by participant) were carried out to explore the temporal stability of these behaviours. A significant main effect of survey month was found for both the distancing and hygiene behaviours (both p < .001; Fig 2). Distancing behaviours showed a notable decline with each subsequent survey (Table 5). By contrast, hygiene behaviours showed an initial decline between the April and May surveys, yet returned to baseline levels by July.

## Sensitivity analysis

The public health directive to stay at home except for essentials was relaxed in most states by the time of the June and July surveys. When 'staying at home' was excluded from the PCA, the pattern of findings was generally similar. One key difference was that the temporal decrease in distancing behaviours was smaller, though still statistically significant. This suggests that the 'staying at home' behaviour was not the sole reason for the observed decrease in distancing behaviours over time (Fig 2, S2 Table).

## Regression modelling of distancing and hygiene components June and July

Higher perceived societal risk, self-efficacy to maintain distancing, and social obligation in April significantly predicted distancing behaviours in June and July, controlling for distancing behaviour in April, health comorbidities and other sociodemographic variables (p's <0.05)

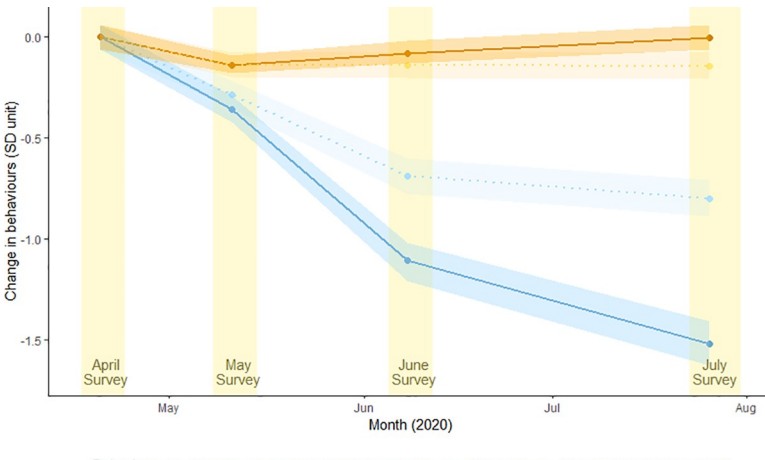

**Fig 2. Change in distancing and hygiene behaviours between April and July, relative to April Survey variance (SD unit).** Units refer to standard deviation of baseline behaviours (hygiene and distancing). The sensitivity analysis did not include the 'staying at home' behaviour in the PCA model.

(Table 6). Personal risk perception in April predicted distancing behaviours in June when community transmission was low (b = -0.65, 95% CI: -1.14, -0.16; p = 0.009), but not July (b = 0.15, 95%CI: -0.36, 0.67; p = 0.55) when new outbreaks were occurring. In contrast, perceived personal control over likelihood of family COVID-19 infection in April predicted distancing behaviours during outbreaks in July (b = 0.26, 95%CI: 0.10, 0.42; p = 0.002) but not low cases in June (b = 0.08, 95% CI: -0.06, 0.23; p = 0.28).

Perceived personal control over likelihood of family COVID-19 infection was the only significant predictor of hygiene behaviours in both June (b = 0.20, 95%CI: 0.11, 0.29; p<0.001) and July (b = 0.20, 95%CI: 0.11, 0.30; p<0.001).

## Discussion

This study found that in a social media sample of participants residing in Australia, there were consistently high levels of seven self-reported COVID prevention behaviours across a four-month period (April to July, 2020). The observed patterns suggest these behaviours can be characterised as either 'distancing' behaviours (staying at home, staying 1.5m away from others, avoiding people with flu-like symptoms, and avoiding shaking hands, hugging or kissing) or 'hygiene' behaviours (washing hands, avoiding touching your face, and cleaning frequently touched surfaces). These two categories of behaviour were moderately correlated but displayed different trajectories over time. Whilst the extent of hygiene behaviours was similar across

**Table 5. Pairwise comparisons between the April (baseline) and subsequent surveys on distancing and hygiene component scores.**

| Pairwise comparisons to April Survey (baseline) | Component 1: Distancing Estimated mean difference (95% CI); p-value | Component 2: Hygiene Estimated mean difference (95% CI); p-value |
| --- | --- | --- |
| May Survey | -0.36 (-0.42, -0.29), p < .001 | -0.14 (-0.18, -0.09), p < .001 |
| June Survey | -1.11 (-1.21, -1.02), p < .001 | -0.08 (-0.13, -0.02), p = .007 |
| July Survey | -1.52 (-1.63, -1.41), p < .001 | -0.003 (-0.06, 0.06), p = .92 |

Values are presented as estimated mean differences from the fixed portion of the linear mixed models, and can be interpreted as standard deviation units.

**Table 6. Multivariable regression modelling of distancing and hygiene behaviours during the June and July Surveys, with baseline responses included as explanatory variables** *.

| Explanatory Variable | Distancing behaviours | | Hygiene behaviours | |
|---|---|---|---|---|
| | June Survey Coefficient (95% CI), p-value | July Survey Coefficient (95% CI), p-value | June Survey Coefficient (95% CI), p-value | July Survey Coefficient (95% CI), p-value |
| **Risk perception** | | | | |
| Societal risk perception^ | 0.06 (0.01, 0.11), p = .026 | 0.08 (0.03, 0.14), p = .003 | 0.01 (-0.03, 0.04), p = .71 | 0.01 (-0.03, 0.04), p = .72 |
| Personal risk perception | -0.65 (-1.14, -0.16), p = .009 | 0.15 (-0.36, 0.67), p = .55 | -0.18 (-0.49, 0.13), p = 0.25 | -0.01 (-0.31, 0.30), p = .97 |
| **Self-efficacy to maintain distancing** | | | | |
| Over a 1-month period: 1 (low) to 7 (high) | 0.21 (0.16, 0.27), p < .001 | - | 0.02 (-0.01, 0.06), p = .19 | - |
| Over a 3-month period: 1 (low) to 7 (high) | - | 0.14 (0.09, 0.19), p < .001 | - | 0.02 (-0.01, 0.05), p = .24 |
| **Consequence of personal actions** | | | | |
| Control over likelihood of personal COVID-19 infection | 0.01 (-0.04, 0.07), p = .58 | -0.01 (-0.07, 0.05), p = .75 | 0.00 (-0.03, 0.03), p = .99 | -0.01 (-0.04, 0.02), p = .56 |
| Control over likelihood of family COVID-19 infection | 0.08 (-0.06, 0.23), p = .26 | 0.26 (0.10, 0.42), p = .002 | 0.20 (0.11, 0.29), p < .001 | 0.20 (0.11, 0.30), p < .001 |
| **Social obligation** | | | | |
| Social obligation | 0.39 (0.17, 0.62), p = .001 | 0.27 (0.04, 0.51), p = .022 | -0.06 (-0.20, 0.07), p = .37 | 0.01 (-0.12, 0.15), p = .83 |

* Models adjusted for baseline (April Survey) behaviours (distancing or hygiene), age, gender, socioeconomic status, residential remoteness, education, residential state, and health comorbidities. Models are adjusted for the corresponding baseline behaviour component (i.e. models for distancing behaviours are adjusted for baseline distancing behaviours but not baseline hygiene behaviours).

^ in the April survey, this question was given without locational framing; for the remaining surveys this was asked specifically within an Australian context. N/A indicates this question was not assessed.

time-points, distancing behaviours decreased over time, with the largest decrease observed in July (equivalent to 1.5 standard deviations below the mean level reported for these behaviours in April).

Distancing behaviours in June and July were predicted by greater perceived risk of COVID-19 to society, feelings of social obligation, and self-efficacy to maintain distancing in April. In contrast, the only variable to consistently predict hygiene behaviours in June and July was belief that distancing could prevent infection of family members. Notably, distancing and hygiene behaviours were not predicted by belief that the respondent could control whether or not they became infected, and personal risk perception was only a significant predictor of one behaviour type (distancing) at one of the two timepoints (June). Together this suggests that in the Australian context, perceptions of community risk and safety may be larger drivers of COVID prevention behaviour than perceptions of personal risk and safety. This is consistent with our findings on vaccine acceptability (based on the same pool of participants) [31], wherein two of the most common reasons for willingness to get a COVID-19 vaccine were 'to protect self and others' and 'to help stop the virus spread.' More broadly, these findings also support behavioural scientists' call for COVID-19 messaging that promotes prosocial behaviour and a collective identity [9, 18, 32].

Specific characteristics of the behaviours investigated in this study may have contributed to relatively stable rates of hygiene behaviours and decreasing distancing behaviours over time. For example, the behaviours categorised as 'hygiene' were also those that were simpler, less disruptive to daily life, and were unlikely to have a strong impact on social interactions. By contrast, the behaviours categorised as 'distancing' behaviours were more complex and clearly placed limits on social interaction. This could explain why, despite consistently strong sentiments of social obligation in our sample, distancing behaviours decreased as external barriers

to social interactions were removed (that is, as national restrictions were eased). Further, this decrease in distancing behaviours continued into July, a time at which the number of daily confirmed cases had increased, when our sample reported elevated perceptions of societal risk, and a strong belief in a social obligation to engage in COVID prevention behaviours.

Our findings show that the concept of behavioural fatigue [22, 23] is simplistic and needs further consideration. Patterns of behaviour over time may depend on factors such as characteristics of the behaviours themselves (e.g. how complex or disruptive they are), the social and physical context (e.g. the extent that people are able to spend time in public areas), and perceived and actual levels of risk in the local environment. This study showed clear evidence that hygiene behaviours were more easily maintained than distancing behaviours, particularly as restrictions eased. As seen in the Australian state of Victoria, reduced distancing behaviours over time can be overcome with enforced restrictions when required and then eased again when cases are under control [33]. A mix of individual behaviour change to prevent outbreaks, and policy change to control outbreaks, has been an effective way to manage COVID-19 in the Australian context where trust in government is high [25]. However, this approach may not transfer to other countries where trust in government and institutions is lower [34].

In this study we observed that self-efficacy, risk perception and social obligation in April were associated with distancing behaviour in June and July. Overall, these findings are consistent with other COVID-19 research on distancing behaviours [12–14]. These findings are also consistent with cross-sectional surveys which report that risk perception and self-efficacy are associated with COVID prevention behaviours, along with trust in government and science [15–21].

Interestingly, our study did not find that self-efficacy, risk perception or social obligation were associated with hygiene behaviours. Whilst we acknowledge that this may reflect an emphasis on distancing in the self-efficacy and social obligation items, it may be worth considering whether other factors better predict hygiene behaviours. For example, as described above, hygiene behaviours are comparatively simple. It is plausible that by June and July, habits had already become established, or that simple environmental cues such as a high availability of hand sanitisers had been sufficient to encourage these behaviours.

## Strengths and limitations

This study has several strengths. Firstly, the findings are contextualised within the Australian experience of the pandemic, which by international standards responded well to the pandemic and may serve as a model for international comparison. By following participants over a period of four months this study is able to provide valuable insight into relative differences in COVID prevention behaviours across three vastly different scenarios: 1) new nationally-imposed restrictions; 2) low and stable daily cases of COVID-19 during eased restrictions; and 3) increasing numbers of daily cases. Secondly, by analysing COVID prevention behaviours collectively, this study conserves statistical power, reduces the Type 1 error rate, and provides a typology of these behaviours that may be useful for future strategic planning.

We also acknowledge this study's limitations. Interpretation of some of these items may have changed over time. For example, the sense of 'distancing' may have changed in Australia between April and July. In April, during national restrictions, government advice on distancing clearly made reference to strangers as well as family and friends. As social venues such as restaurants have opened up, distancing is more likely to refer to strangers and implicitly excludes close friends or family. If this is the case and the 'meaning' of distancing has become less conservative over time, this current analysis may underestimate the decrease in distancing behaviours over time. For example, people may apply the 1.5 metre rule to strangers in public

but not when visiting friends or extended family in other homes, a common setting for COVID-19 clusters in Australia. Additionally, self-reported data may be subject to recall and desirability biases that reduce accuracy. However, self-reported behavioural measures have practical advantages and are widely used in COVID-19 research [3–7, 12–20].

We also acknowledge that recruitment via social media has resulted in a sample that is younger, more educated and has a higher proportion of women than the Australian population. Results may not generalise to older, less educated samples with a higher proportion of men [35]. Similarly, although the sample reflected national distributions in remoteness, a larger proportion of the sample resided in NSW (51%) compared to the population (32%). Given that NSW had more cases than other states/territories excluding Victoria, and given that policies differed by state, findings in this study may not apply as strongly to other states with fewer local cases. The regression models presented in the results do control for these factors but may not entirely remove this artifact. This sample consented to take part in a longitudinal study on COVID-19 and so may be more likely to have an active interest in COVID-19. As such, participants in this sample may be more engaged in government recommendations for prevention behaviours. Despite these limitations, the extent of COVID prevention behaviours reported in this paper mirrors the high estimates reported in nationally representative surveys [3–7], and the sample is representative on other factors such as remoteness and number of chronic conditions.

## Future directions

As we observed different patterns over time for hygiene and distancing behaviours, future research should carefully consider which behaviours are included in analyses of COVID prevention behaviours, and whether these behaviours are analysed collectively or independently. For policy makers, our results suggest that environmental cues and messages may be sufficient to maintain new habits for simple hygiene behaviours, but enforcement may be required for more complex social behaviours like distancing as this appears to be more difficult for individuals to regulate and maintain. Future research could investigate the mechanisms by which environmental and psychological factors facilitate or impede habit formation for COVID-19 prevention behaviours and the effectiveness of enforcement strategies. Future research on COVID-19 behaviours should also consider diverse groups; currently we are recruiting a survey sample across ten language groups in Greater Western Sydney.

The findings from this study also highlight potential strategies for public health messaging. We observed that externally focused factors such as perceived societal risk, the ability to protect others such as family members, and social obligation, were more closely associated with distancing behaviour than personal considerations. Future research could investigate how sentiments regarding social obligation can be bolstered, and whether framing messages in terms of community benefit is a more effective strategy for encouraging COVID prevention behaviours. For example, one study has found that evoking a highly positive emotional response using prosocial persuasive language was more effective at encouraging self-isolation behaviours than messages that emphasised the threat of COVID-19 [36].

## Conclusion

In this Australian online social media sample, participants reported consistently high levels of seven COVID prevention behaviours between April and July, 2020. Distancing behaviours decreased over time, whereas hygiene behaviours remained relatively stable. Along with self-efficacy, factors relating to others and the community at large were more likely to predict behaviours than personal considerations. This study highlights the importance of considering

how patterns of COVID prevention behaviours change over time, and suggests that different policy approaches may be needed for different behavioural categories.

## Supporting information

**S1 Table. Descriptive characteristics of analysis sample (N = 1843) and full sample invited for follow-up (N = 3216) at baseline (April).**
(DOCX)

**S2 Table. Sensitivity analyses of pairwise comparisons between the April (baseline) and subsequent surveys on distancing and hygiene component scores (i.e., 'stay at home' behaviour not included in the PCA).** Values are presented as estimated mean differences from the fixed portion of the linear mixed models, and can be interpreted as standard deviation units.
(DOCX)

**S1 Data. Supporting data.**
(XLSX)

## Author Contributions

**Conceptualization:** Julie Ayre, Erin Cvejic, Kirsten McCaffery, Tessa Copp, Samuel Cornell, Rachael H. Dodd, Kristen Pickles, Carys Batcup, Jennifer M. J. Isautier, Brooke Nickel, Thomas Dakin, Carissa Bonner.

**Data curation:** Julie Ayre, Carys Batcup.

**Formal analysis:** Julie Ayre, Erin Cvejic.

**Funding acquisition:** Kirsten McCaffery.

**Investigation:** Julie Ayre, Carys Batcup.

**Methodology:** Julie Ayre, Erin Cvejic, Kirsten McCaffery, Tessa Copp, Samuel Cornell, Rachael H. Dodd, Kristen Pickles, Carys Batcup, Jennifer M. J. Isautier, Brooke Nickel, Thomas Dakin, Carissa Bonner.

**Project administration:** Kirsten McCaffery.

**Resources:** Julie Ayre, Erin Cvejic, Carys Batcup.

**Software:** Julie Ayre, Erin Cvejic, Carys Batcup.

**Supervision:** Kirsten McCaffery.

**Validation:** Julie Ayre, Erin Cvejic.

**Visualization:** Julie Ayre.

**Writing – original draft:** Julie Ayre, Erin Cvejic.

**Writing – review & editing:** Julie Ayre, Erin Cvejic, Kirsten McCaffery, Tessa Copp, Samuel Cornell, Rachael H. Dodd, Kristen Pickles, Carys Batcup, Jennifer M. J. Isautier, Brooke Nickel, Thomas Dakin, Carissa Bonner.

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
