## [Decision Letter · Decision Letter 0]

28 Apr 2021

PONE-D-21-07536

Contextualising COVID-19 prevention behaviour over time in Australia: Patterns and long-term predictors from April to July 2020

PLOS ONE

Dear Dr. Ayre,

Thank you for submitting your manuscript to PLOS ONE. After careful consideration, we feel that it has merit but does not fully meet PLOS ONE’s publication criteria as it currently stands. Therefore, we invite you to submit a revised version of the manuscript that addresses the points raised during the review process.

The authors have presented a well-written piece of evidence regarding COVID-preventive behaviors in Australia over the months of April to July, 2020. They have attempted to describe the changing pattern of these behvaiors over time and their predictors. The comments and concerns are appended below. Please address all the comments on a point-by-point basis.

We look forward to receiving your revised manuscript.

Kind regards,

Arista Lahiri, MBBS, MD

Academic Editor

PLOS ONE

Journal Requirements:

Additional Editor Comments:

The authors have presented a well-written piece of evidence regarding COVID-preventive behaviors in Australia over the months of April to July, 2020. They have attempted to describe the changing pattern of these behvaiors over time and also sought to understand the role of certain behavioral predictors. However, there are certain concerns and the article needs further clarification on the following issues:

1. Sampling of participants and representativeness of the sample. The authors must also revise their title, because I am afraid the findings do not represent the whole Australian population, rather only a digitally advanced group of people were studied. This part needs clarification and requisite modifications.

2. Considering the study-tool to be a non-standard one, detailed validity and reliability information is needed.

3. The non-response and drop out in the study appears to be high. Did the authors compare the basic characteristics between the responders and non-responders?

4. Among the respondents in different months, were they the same respondents as from previous months, or were there any new respondents? Authors should make this part clearer. Because based on this the analytical strategy may change.

5. The authors have taken into account several COVID-preventive behaviors and based on the responses they have grouped the behaviors in two major themes i.e., distancing and hygiene. Now, in the regression models, the authors have developed separate models for every month of survey. The variables in the regression analyses I feel, may suffer from endogeneity. Authors may please look into it.

6. In order to understand trend or patterns it might have been better to develop a regression model with time as a predictor. In the current study, a multi-level model can be used to make the findings more robust, crisp, and comprehensive. Though I have reservation regarding use of logistic regression only, as the method of analysis in a study where the authors aimed to see the patterns and the predictors, still the use of other applicable or rather more appropriate models is dependent upon the data collection and sampling itself (refer point 4).

7. In abstract authors may please follow this format: background, objectives, methods, results, conclusion.

Reviewers' comments:

Reviewer's Responses to Questions

**Comments to the Author**

1. Is the manuscript technically sound, and do the data support the conclusions?

Reviewer #1: Yes

Reviewer #2: Yes

Reviewer #3: Yes

2. Has the statistical analysis been performed appropriately and rigorously? 

Reviewer #1: Yes

Reviewer #2: Yes

Reviewer #3: Yes

3. Have the authors made all data underlying the findings in their manuscript fully available?

Reviewer #1: Yes

Reviewer #2: No

Reviewer #3: Yes

4. Is the manuscript presented in an intelligible fashion and written in standard English?

Reviewer #1: Yes

Reviewer #2: Yes

Reviewer #3: Yes

5. Review Comments to the Author

Reviewer #1: This study highlights the importance of considering how patterns of COVID prevention behaviours change over time, and suggests that different policy approaches especially risk communication approaches, may be needed for different behavioural categories in different stages of the pandemic.

The data has been elaborately analysed and presented. Appropriate statistical analysis been performed supporting the conclusions drawn.

The manuscript is written in a lucid and unambiguous form which is very comprehensible.

As a future research direction, the authors could consider analysis of complexity of performance of the seven precautionary behaviors along with their contextual facilitators and barriers.

Reviewer #2: Overall paper is good.

1. Sample size seems low

2. Citation format issue on page 15 Betsch, Schmid, Heinemeier, Korn, Holtmann and Böhm (30)

3. NSW and the Australian Capital Territory is over represented in collected data otherwise sampling is very good.

4. On page 22, figure 2 missing on page but included later

5. Only overall behavior over time is shown in graphs, not behavior/perceptions separated by demographics.

6. Paper doesn't point out if there were other efforts made to get data from people outside Facebook(email marketing, ads on Youtube, etc) but does point out that sampling methods could affect results.

Reviewer #3: I have 2 main concerns with the work:

1) accuracy of self reporting - i am unsure of how accurate self reporting for these behaviours really is as an assessment tool

2) sample - considering that Australia has roughly 25.3 million people, a sample size of 1834 does not seem representative of the overall population. Moreover the sample encompasses a majority of highly educated women, which again reinforces that the sample is very selective compared to the national australian estimates.

The article is well written but i question the significance of the results based on such a small and non-representative sample and I because of that I don't think these results can be extrapolated to the australian population at large.

6. PLOS authors have the option to publish the peer review history of their article (what does this mean?). If published, this will include your full peer review and any attached files.

Reviewer #1: **Yes: **Madhumita Dobe

Reviewer #2: No

Reviewer #3: No

---

## [Author Response · Author response to Decision Letter 0]

25 May 2021

Please see attached 'response to reviewer comments' letter.

---

## [Decision Letter · Decision Letter 1]

16 Jun 2021

Contextualising COVID-19 prevention behaviour over time in Australia: Patterns and long-term predictors from April to July 2020 in an online social media sample

PONE-D-21-07536R1

Dear Dr. Ayre,

We’re pleased to inform you that your manuscript has been judged scientifically suitable for publication and will be formally accepted for publication once it meets all outstanding technical requirements.

Kind regards,

Arista Lahiri

Academic Editor

PLOS ONE

Additional Editor Comments (optional):

The authors have put in great amount of work. Upon revision the article is now scientifically Acceptable.

Reviewers' comments:

Reviewer's Responses to Questions

**Comments to the Author**

1. If the authors have adequately addressed your comments raised in a previous round of review and you feel that this manuscript is now acceptable for publication, you may indicate that here to bypass the “Comments to the Author” section, enter your conflict of interest statement in the “Confidential to Editor” section, and submit your "Accept" recommendation.

Reviewer #1: All comments have been addressed

Reviewer #2: All comments have been addressed

2. Is the manuscript technically sound, and do the data support the conclusions?

Reviewer #1: Yes

Reviewer #2: Yes

3. Has the statistical analysis been performed appropriately and rigorously? 

Reviewer #1: Yes

Reviewer #2: Yes

4. Have the authors made all data underlying the findings in their manuscript fully available?

Reviewer #1: Yes

Reviewer #2: No

5. Is the manuscript presented in an intelligible fashion and written in standard English?

Reviewer #1: Yes

Reviewer #2: Yes

6. Review Comments to the Author

Reviewer #1: (No Response)

Reviewer #2: The comments made previously have been addressed, including mentioning the limitations of the data collection process and related issues regarding the relatively small sample.

7. PLOS authors have the option to publish the peer review history of their article (what does this mean?). If published, this will include your full peer review and any attached files.

Reviewer #1: **Yes: **Madhumita Dobe

Reviewer #2: **Yes: **Muhammad Sohaib Arif

---

## [Editor Report · Acceptance letter]

21 Jun 2021

PONE-D-21-07536R1 

Contextualising COVID-19 prevention behaviour over time in Australia: Patterns and long-term predictors from April to July 2020 in an online social media sample 

Dear Dr. Ayre:

I'm pleased to inform you that your manuscript has been deemed suitable for publication in PLOS ONE. Congratulations! Your manuscript is now with our production department. 

Kind regards, 

on behalf of

Dr. Arista Lahiri 

Academic Editor

PLOS ONE